

# An adaptive singular spectrum analysis method for extracting brain rhythms of electroencephalography

Hai Hu[1], Shengxin Guo[2], Ran Liu[2] and Peng Wang[1]

[1] State Key Laboratory of Precision Measurement Technology and Instruments, Tsinghua University, Beijing, China
[2] Department of Biomedical Engineering, School of Medicine, Tsinghua University, Beijing, China

## ABSTRACT

Artifacts removal and rhythms extraction from electroencephalography (EEG) signals are important for portable and wearable EEG recording devices. Incorporating a novel grouping rule, we proposed an adaptive singular spectrum analysis (SSA) method for artifacts removal and rhythms extraction. Based on the EEG signal amplitude, the grouping rule determines adaptively the first one or two SSA reconstructed components as artifacts and removes them. The remaining reconstructed components are then grouped based on their peak frequencies in the Fourier transform to extract the desired rhythms. The grouping rule thus enables SSA to be adaptive to EEG signals containing different levels of artifacts and rhythms. The simulated EEG data based on the Markov Process Amplitude (MPA) EEG model and the experimental EEG data in the eyes-open and eyes-closed states were used to verify the adaptive SSA method. Results showed a better performance in artifacts removal and rhythms extraction, compared with the wavelet decomposition (WDec) and another two recently reported SSA methods. Features of the extracted alpha rhythms using adaptive SSA were calculated to distinguish between the eyes-open and eyes-closed states. Results showed a higher accuracy (95.8%) than those of the WDec method (79.2%) and the infinite impulse response (IIR) filtering method (83.3%).

## INTRODUCTION

Electroencephalography (EEG) is a measurable voltage resulting from electrical activity of the brain neurons (*Niedermeyer & Da Silva, 2005*). Spontaneous EEG consists of several rhythms of different frequencies: delta (1–4 Hz), theta (4–8 Hz), alpha (8–13 Hz), beta (13–30 Hz) and gamma (>30 Hz), each containing information about different brain activity. For example, the alpha rhythm reflects attention demands and the beta rhythm reflects emotional and cognitive processes (*Rowland, Meile & Nicolaidis, 1985*). Because of its feasibility and convenience, EEG has been widely studied for the brain physiological states monitoring (*Jones et al., 2014*; *Ko, Yang & Sim, 2009*; *Ng & Chan, 2005*).

Rhythms extraction from the EEG signals is important for portable and wearable EEG recording devices, which have attracted much attention in recent studies (*Gargiulo et*

Corresponding author
Peng Wang,
peng@mail.tsinghua.edu.cn

*al., 2008; Van Bavel et al., 2008). Chi & Cauwenberghs (2010)* developed a system using wireless non-contact EEG electrodes to collect EEG signals for the alpha rhythm extraction. Ranjit et al. utilized changes in the EEG alpha rhythm during eye closure as a switch for electrical devices used for severely impaired people (*Thuraisingham et al., 2007*). In addition, the EEG alpha rhythm was also used as a measure of resting-state arousal and activation (*Barry et al., 2007*). However, the EEG signal is always contaminated by artifacts, including electrooculogram (EOG), electromyography (EMG), baseline drift and stochastic noise, which interfere the rhythms extraction (*Azarbad et al., 2014; Daly et al., 2013; Teixeira et al., 2006*).

To extract the desired rhythms accurately from the interfered EEG signals, various methods have been proposed for artifacts removal (*Azarbad et al., 2014; Daly et al., 2013; Teixeira et al., 2006*). In multi-channel EEG recording, visual inspection is useful for artifacts removal. However, this method is not applicable for portable or wearable devices with single-channel EEG recording (*Nunez et al., 1999*). Therefore, many efforts have been paid on algorithm for artifacts removal (*Teixeira et al., 2006*). *He, Wilson & Russell (2004)* used regression analysis (RA) and adaptive filtering (AF) techniques for EOG artifact removal. These methods required a separately recorded EOG signal as a reference, which was however always contaminated. *Wallstrom et al. (2004)* applied the independent component analysis (ICA) method to automatically remove the ocular artifacts in EEG signals. However it distorted the EEG signals in the range of 5–20 Hz. Azami et al. used the wavelet transform (WT) method for artifacts removal. However this method was relatively slow, and could not separate components overlapping in time-frequency space (*Azami & Sanei, 2012; Azami & Sanei, 2014*).

SSA is another powerful method for time series analysis. It enables separation of different sources even overlapping in time-frequency space, and has been recently applied for EEG artifacts removal and rhythms extraction. *Maddirala & Shaik (2016b)* proposed a new grouping criterion in SSA to construct the reference signal for EOG artifact removal. Based on the local mobility of the eigenvectors, Maddirala et al. then proposed another grouping criterion to remove the motion artifact, which performed better than the existing methods (*Maddirala & Shaik, 2016a*). *Mohammadi et al. (2015)* and *Mohammadi et al. (2016)* proposed a SSA method with a new grouping criterion based on the eigenvalue pairs to extract the main rhythms from sleep EEG signals. Akar et al. proposed a wavelet decomposition-SSA based method for noise removal and desired components extraction from EEG signals. It was successfully applied for schizophrenics' brain dynamics analysis (*Akar et al., 2015*). In the SSA method, the grouping rule is important for SSA reconstruction. However, there is no general grouping rule. For a specific study, it depends on the research target, the types of signals and noise involved (*Azami & Sanei, 2014; Kouchaki, 2014; Sanei, Lee & Abolghasemi, 2012*).

In this paper, we proposed a novel grouping rule for SSA reconstruction to remove artifacts and extract rhythms from EEG signals. EEG signals were processed by SSA to obtain a series of reconstructed components The first one or two reconstructed components determined by the signal amplitude were then regarded as artifacts and removed. Finally, the reconstructed components were grouped, based on their peak frequencies in the

Fourier transform, to extract the specific EEG rhythms. Proof-of-concept experiments were performed to verify the proposed adaptive SSA method in extracting the EEG alpha rhythm and distinguishing between the eyes-open and eyes-closed states.

## METHODS

### Adaptive SSA method

SSA consists of two stages: decomposition and reconstruction. Decomposition involves time-delay embedding called Takens' theorem (*Takens, 1981*), followed by singular value decomposition (SVD) (*Mees, Rapp & Jennings, 1987*). Reconstruction involves grouping and diagonal averaging (*Vautard, Yiou & Ghil, 1992*). In the time-delay embedding step, the single-channel EEG time series $\mathbf{s} = (s_1, s_2, \ldots, s_N)^T$, superscript $T$ denoting the transpose of a vector is mapped onto a multidimensional trajectory matrix $\mathbf{X}$ using a sliding window

$$\mathbf{X} = (\mathbf{S}_1, \mathbf{S}_2, \ldots, \mathbf{S}_K) = \begin{pmatrix} s_1 & s_2 & \cdots & s_K \\ s_2 & s_3 & \cdots & s_{K+1} \\ \vdots & \vdots & \ddots & \vdots \\ s_L & s_{L+1} & \cdots & s_N \end{pmatrix} \tag{1}$$

where $L$ denotes the window length (or embedding dimension), $K = N - L + 1$, and $\mathbf{S}_i (1 \leq i \leq k)$ denotes the lagged vector. Next, the SVD of the matrix $\mathbf{X}$ is computed as:

$$\mathbf{X} = \sum_{i=1}^{L} \mathbf{X}_i = \sum_{i=1}^{L} \sqrt{\lambda_i} \mathbf{v}_i \mathbf{p}_i^T \tag{2}$$

where $\mathbf{X}_i$ denotes the elementary matrice, $\lambda_i$ denotes the eigenvalue of covariance matrix $\mathbf{C} = \mathbf{X}\mathbf{X}^T$ in the decreasing order of magnitude ($\lambda_1 \geq \lambda_2 \geq \ldots \geq \lambda_L \geq 0$), $\mathbf{v}_i$ denotes the corresponding eigenvector, and $\mathbf{p}_i = \mathbf{X}^T \mathbf{v}_i / \sqrt{\lambda_i}$.

Then, the diagonal averaging step reconstructs several time series from the corresponding elementary matrices. The reconstructed time series are generally called reconstructed components (RCs). Finally, based on the proposed adaptive grouping rule, the RCs are grouped for artifacts removal and rhythms extraction.

### *The adaptive grouping rule*

With the SSA treatment the original EEG time series are decomposed into a set of RCs. The first several RCs dominate the trend of the EEG time series, which is represented by the large artifacts (*Teixeira et al., 2006*). Here, the first one or two RCs, determined by the EEG amplitude, are grouped as artifacts. When the amplitude is large, indicating a high level of artifacts, the first two RCs are grouped. Otherwise, the first RC is grouped:

$$Artifacts = \begin{cases} RC_1 + RC_2 & \max(\mathbf{s}) > V_0 \\ RC_1 & \max(\mathbf{s}) < V_0 \end{cases} \tag{3}$$

where $\max(\mathbf{s})$ denotes the EEG amplitude $V_0$ is a threshold. Since the amplitude of the spontaneous EEG without artifacts is usually below 100 μV (*Ng & Chan, 2005*), the threshold was set as $V_0 = 200$ μV.
After artifacts removal, the rhythms are then extracted from the EEG time series. Rhythms are the oscillatory components of EEG time series, including, but not limited to, the periodic components. In order to extract the EEG rhythms, the RCs are then divided into two groups: the periodic and non-periodic components. Firstly, the periodic components are extracted. Generally, a periodic time series will be factorised into some eigenvalue pairs with similar amplitude using SSA (*Vautard, Yiou & Ghil, 1992*). So, RCs with similar eigenvalues belong to the same periodic component. They are summed up as a periodic component (PC). The similarity of the eigenvalues for periodic component extraction is determined by the following criterion:

$$|1 - \frac{\lambda_j}{\lambda_i}| < 0.05. \tag{4}$$

Then, these PCs and the remaining non-periodic RCs are grouped. Each PC and RC will fall into a narrow frequency band, when the window length $L$ is large enough (*Kouchaki, 2014*; *Mohammadi et al., 2015*; *Sanei, Ghodsi & Hassani, 2011*). So, the peak frequency in the Fourier transform can be used to represent the frequency range of the PC and RC.

$$f_{\max} = \underset{f}{\text{argmax}}\{abs[FFT(RC)]\} \tag{5}$$

where $f_{\max}$ is the peak frequency, $FFT(RC)$ is the fast Fourier transform of the RC or PC. RCs and PCs with the peak frequencies in the same rhythm band are finally clustered into the same group, which constitutes the brain rhythm.

The pseudo-code of the adaptive SSA method is shown in Fig. 1.

Window length $L$ is selected based on the lowest frequency of interest ($L \geq f_s/f_l$) to capture at least one period of the expected component (*James & Lowe, 2003*). In this paper, the window length $L$ is set to be 40. After extracting the desired rhythms, the features of the rhythms are obtained for each EEG signal. The mean, standard deviation (SD), power and power ratio of different rhythms are usually selected as the features (*Mohammadi et al., 2015*). In this study, the power ($P = \sum V_\alpha^2 / N$) of the alpha rhythm were selected.

## Data source

Simulated EEG data with known parameters and experimental EEG data were used to verify the validity of the adaptive SSA method in artifacts removal and alpha rhythm extraction.

### Simulated EEG data

Simulated EEG data consisted of two parts: the spontaneous EEG and the artifacts. The spontaneous EEG has two major characteristics: rhythmic oscillations and randomness. The artifacts mainly consist of EOG, baseline drift and white Gaussian noise.

The spontaneous EEG was simulated based on the Markov Process Amplitude (MPA) EEG model developed by *Nishida, Nakamura & Shibasaki (1986)*. In the MPA EEG model, the EEG rhythmic oscillations were represented by sinusoidal waves, and the EEG randomness was represented by the stochastic process amplitude of the first-order Markov process (*Al-Nashash et al., 2004*; *Bai et al., 2000*; *Bai et al., 2001*). The spontaneous

---

**Algorithm 1** Adaptive SSA Method for Alpha Rhythms Extraction

**input** **s**: single-channel EEG time series

**output C**: corrected EEG time series after artifacts removal

      **α**: extracted alpha rhythm

**procedures** Artifacts removal and rhythms extraction

    (1) RCs: apply the SSA on **s** with the window length $L$=40 and derive a set of RCs by

       Eq.1-2.

    (2) **C**: derive the artifacts (**A**) by Eq.3, and $\mathbf{C} = \mathbf{s} - \mathbf{A}$

    (3) PCs: group the RCs and derive periodic components (PCs) by Eq.4

    (4) **α**: group the PCs and the non- periodic RCs to derive alpha rhythm by Eq.5

**return C**, **α**

**Figure 1** **The pseudo-code of the adaptive SSA method.**

EEG was generated by a combination of $K$ different oscillations,

$$s(n\Delta t) = \sum_{i=1}^{K} a_i(n\Delta t)\sin(2\pi m_i n\Delta t + \theta_i) \tag{6}$$

where $n$ is the sample number, $\Delta t$ is the time interval, $K$ is the number of rhythms, $m$ is the dominant frequency, $\theta$ is an arbitrary value representing the initial phase, $a$ is the rhythmic amplitude obtained from a first-order Gauss–Markov process

$$a_i[(n+1)\Delta t] = \gamma_i a_i(n\Delta t) + \xi_i(n\Delta t) \tag{7}$$

where $\gamma$ is the coefficient of the first-order Markov process, $\xi$ is a random increment of Gaussian distribution with zero mean and variance $\sigma$.

Figure 2 shows the procedures of the spontaneous EEG simulation based on the MPA EEG model. Firstly, the power spectrum of a real EEG was calculated, as shown in Fig. 2A. In order to achieve the maximum likelihood with respect to the power spectrum of the real EEG, parameters of the MPA EEG model were determined in the frequency domain (shown in Table 1). Then, four oscillations representing the brain rhythms (delta (1–4 Hz), theta (4–8 Hz), alpha (8–13 Hz) and beta (13–30 Hz)) were generated based on the determined parameters, as shown in Fig. 2B. Finally, the spontaneous EEG was generated by a combination of the four rhythms as shown in Fig. 2C. The simulated spontaneous EEG lasted for 8 s with the interval $\Delta t$ of 5 ms. The detailed mathematical description of the spontaneous EEG simulation is shown in the 'Appendix' (Markov Process Amplitude (MPA) EEG model).
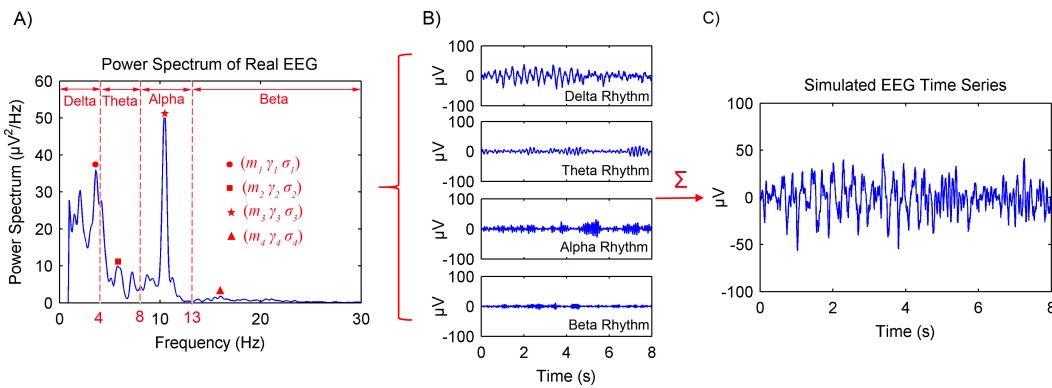

**Figure 2** **Procedures of the spontaneous EEG simulation based on the MPA EEG model.** (A) The power spectrum of a real EEG. Parameters of the model were determined based on the power spectrum. (B) The generated four rhythms: delta, theta, alpha and beta. (C) The simulated spontaneous EEG generated by a combination of the four rhythms.

**Table 1** **Parameters of the simulated EEG time series.**

| | Symbol | Value | Comments |
|---|---|---|---|
| | $m_1$ (Hz) | 3.61 | |
| | $\sigma_1^{\xi}$ | 3.86 | Delta rhythm |
| | $\gamma_1$ | 0.97 | |
| | $m_2$ (Hz) | 5.76 | |
| | $\sigma_2^{\xi}$ | 1.23 | Theta rhythm |
| Spontaneous EEG | $\gamma_2$ | 0.99 | |
| | $m_3$ (Hz) | 10.45 | |
| | $\sigma_2^{\xi}$ | 1.57 | Alpha rhythm |
| | $\gamma_3$ | 0.99 | |
| | $m_4$ (Hz) | 16.02 | |
| | $\sigma_4^{\xi}$ | 0.92 | Beta rhythm |
| | $\gamma_4$ | 0.98 | |
| | $V_{EOG}$ (μV) | 400/100 | Amplitude of EOG |
| | $T_{EOG}$ (s) | 3 | Period of EOG |
| Artifacts | $PW_{EOG}$ (s) | 0.3 | Pulse width of EOG |
| | $V_{BL}$ (μV) | 20 | Amplitude of baseline drift |
| | $f_{BL}$ (Hz) | 0.5 | Frequency of baseline drift |

The artifacts consist of EOG, baseline drift and white Gaussian noise. EOG is the main artifact and is caused by eye blinks and ocular movement. It is characterized by large amplitude, low-frequency electro-potential shift. Baseline drift originates from the head or body movement and is characterized by low-frequency electro-potential shift. White Gaussian noise was used to represent the measurement noise.

Generally, about 15–20 eye blinks, each lasting 0.3–0.4 s will be done in 60 s when the subject is relaxed. Based on the characters, a triangular pulse with the period of 3 s and the pulse width of 0.3 s was chosen to simulate the EOG artifact. A sinusoidal function with amplitude of 20 μV and frequency of 0.5 Hz was used to simulate baseline drift.

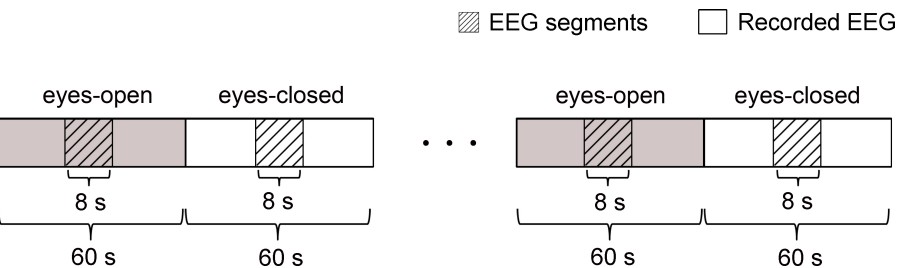

**Figure 3** **Schematic of the recorded EEG data 4 times of alternating periods of 60 s eyes open followed by 60 s eyes closed.** The desired EEG segments were cut off from every period of the eyes-open and eyes-closed states. Each segment was in the middle of each period and last for 8 s.

Parameters used in the EEG simulation are shown in Table 1. The data simulation and treatment were carried out using MATLAB version 7.9.0 on a personal computer with Intel(R) Core(TM) i5-4590 processor, 8 GB RAM and Windows 7 operating system.

### Experimental EEG data

The experiments were approved with a protocol (NO. 20170010) by the Institutional Review Board of Tsinghua University. One male and two female aged 20–25 years participated in the experiments. These subjects were required to abstain from psychoactive substances for at least 4 h prior to experiments. Experiments were carried out with the subject sit on a comfortable chair in a room with normal lightness. The MP36 data acquisition and analysis system (BIOPAC Systems, Inc., Goleta, CA, USA) with a three electrodes system was used to acquire the EEG data. The Ag/AgCl electrode (Wuxi Sichiray Technology Co. Ltd, Wuxi, Jiangsu, China) flushed with conductive gel was attached to the scalp over the frontal region as the recording electrode. The other two electrodes were attached to the earlobe and mastoid, serving as a ground and a reference, respectively.

The experimental procedures were as follows. Initially, the subject closed eyes in relaxed state for 10 min. After that, the subject opened eyes and visually fixated on a small cross displayed on a computer screen in front of him/her. Meanwhile, the EEG data started to be recorded with the sampling rate of 200 Hz. Informed by the recorder, the subject then began 4 times of alternating periods of 60 s eyes open followed by 60 s eyes closed. The desired EEG segments were cut off from every period of the eyes-open and eyes-closed states. As shown in Fig. 3, each segment last for 8 s and was in the middle of each period. For each subject, the experiment was repeated three times in three separate days. Totally, 24 segments were obtained for each subject.

## RESULTS AND DISCUSSION

### Artifacts removal

#### Simulated EEG data

Simulated EEG data based on the MPA EEG model was used to verify the artifacts removal using the adaptive SSA. Representative results are shown in Fig. 4. When the EEG was contaminated with large artifacts (the EOG amplitude was 400 $\mu$V ), as shown in Fig. 4A, the amplitude of the EEG signal was higher than the threshold ($V_0 = 200$ $\mu$V). The first two

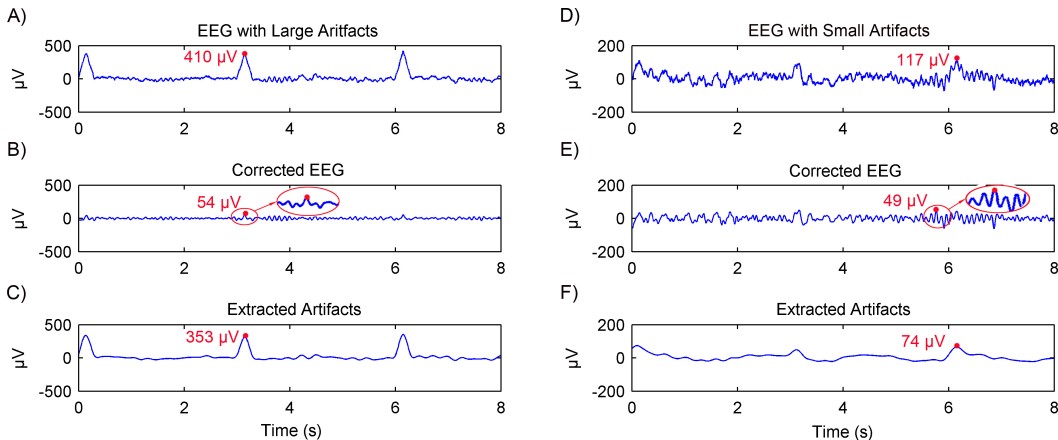

**Figure 4** **Results of artifacts removal from simulated EEG using the adaptive SSA.** EEG contaminated with artifacts (A), the corrected EEG (B) and removed artifacts (C), in the case of large artifacts. The amplitude of the EOG artifact is 400 μV. EEG contaminated with artifacts (D), the corrected EEG (E) and removed artifacts (F), in the case of small artifacts. The amplitude of the EOG artifact is 100 μV.

RCs were grouped as artifacts and removed. The corrected EEG and removed artifacts are shown in Figs. 4B and 4C, respectively. It can be seen that the amplitude of the corrected EEG was 54 μV, close to that of the simulated spontaneous EEG as shown in Fig. 2C. When the EEG was contaminated with small artifacts (the EOG amplitude was 100 μV), as shown in Fig. 4D, the first RC was grouped as artifacts and removed. The amplitude of the corrected EEG was 49 μV, close to that of the spontaneous EEG as well. Therefore, it could be concluded that the large or small artifacts could be removed adaptively by the adaptive SSA.

### Experimental EEG data

The artifacts removal using the adaptive SSA was then further tested on experimental EEG data. Another recently reported SSA method (*Maddirala & Shaik, 2016b*), which will be called SSA 1# in this paper, was used for comparison. The SSA 1# used a novel grouping criterion based on the eigenvectors' local mobility, which is a signal complexity measure, to remove motion artifacts. The comparison results are shown in Fig. 5. For nine of the total 24 segments, the corrected EEG using the adaptive SSA was the same as that using the SSA 1#. Representative results are shown in Figs. 5B and 5C. Furthermore, it can be seen from Fig. 5D that their power spectrums overlapped completely. For the other 15 of the total 24 segments, the amplitude of the corrected EEG using the adaptive SSA was higher than that of the corrected EEG using the SSA 1#, as shown in Figs. 5F and 5G, respectively. It can be seen from Fig. 5H that the SSA 1# removed more artifacts than the adaptive SSA. The excess removed artifacts were in the frequency of 3–5 Hz, which was within the EEG frequency band and should not be removed. Therefore, the corrected EEG using the adaptive SSA was more complete and accurate than that using the SSA 1#. It verifies that the adaptive SSA has a similar, even better, artifacts removal effect.

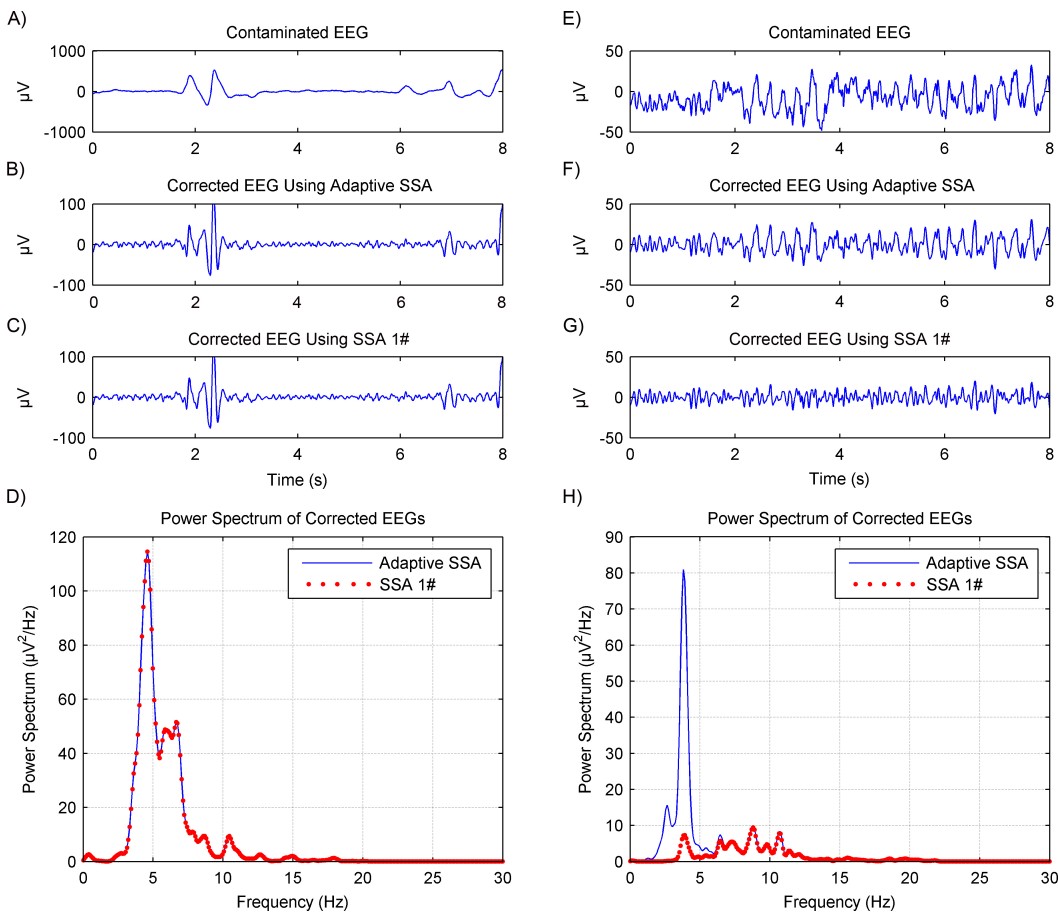

**Figure 5  Comparison results of artifacts removal between using the adaptive SSA and SSA 1#.** The contaminated EEG (A), corrected EEG using the adaptive SSA (B) and SSA 1# (C), and power spectrums of the corrected EEG (D), in the case of producing the same results using the adaptive SSA and SSA 1# The contaminated EEG (E), corrected EEG using the adaptive SSA (F) and SSA 1# (G), and power spectrums of the corrected EEG (H), in the case of producing different results using the adaptive SSA and SSA 1#.

## Alpha rhythms extraction
### *Simulated EEG data*
Simulated EEG data based on the MPA model was then used to verify the rhythms extraction using the adaptive SSA. Results are shown in Fig. 6. The extracted alpha rhythm using the adaptive SSA is shown in Fig. 6A. It was similar with the simulated alpha rhythm (shown in Fig. 6B). To further examine the rhythms extraction, their power spectrums were calculated (shown in Fig. 6C). The power spectrum of the extracted alpha rhythm was in good consistence with that of the simulated alpha rhythm. It verifies the validity of the adaptive SSA in rhythms extraction.

### *Experimental EEG data*
The rhythms extraction using the adaptive SSA was then further tested on experimental EEG data. After artifacts removal, the EEG alpha rhythms in the eyes-open and eyes-closed states were extracted. Figs. 7A and 7B shows the representative alpha rhythm extracted

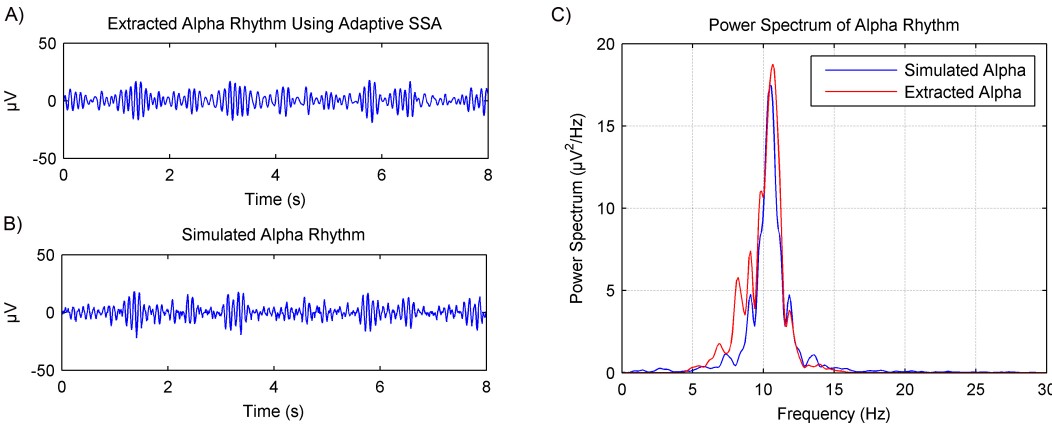

**Figure 6** **Results of the alpha rhythm extraction from simulated EEG data using the adaptive SSA.** (A) The extracted alpha rhythm using the adaptive SSA. (B) The simulated alpha rhythm. (C) Power spectrums of the alpha rhythms in (A) and (B).

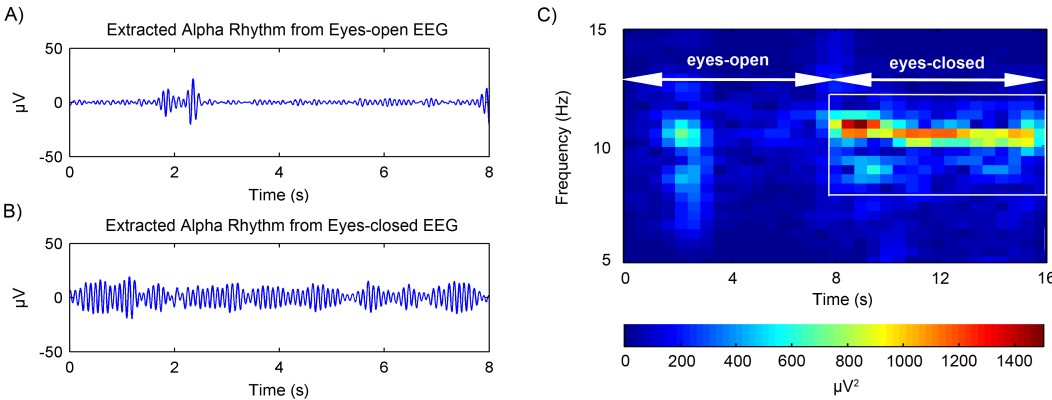

**Figure 7** **Results of the alpha rhythm extraction in (A) eyes-open state and (B) eyes-closed state, respectively. (C) The spectrogram of the alpha rhythm.**

from EEG signals in the eyes-open and eyes-closed states, respectively. It is clear that the alpha rhythm in the eyes-open state was weaker than that in the eyes-closed state. Actually, the power value of the extracted alpha rhythm in the eyes-open state was 9.84 $\mu V^2$; while in the eyes-closed state the power value was 50.52 $\mu V^2$. Figure 7C illustrates the alpha rhythm spectrogram, which is the square of the rhythm amplitude as a function of frequency. It presents an obvious difference between the eyes-open and eyes-closed states.

In order to verify the rhythms extraction using the adaptive SSA, another recently reported SSA method (*Mohammadi et al., 2016*) and wavelet decomposition method (*Akar et al., 2015*), which will be called SSA 2# and WDec respectively in this paper, were used for comparison. The comparison results are shown in Fig. 8. It can be seen from Figs. 8A and 8D that, in the eyes-open state, the extracted alpha rhythm using the adaptive SSA was of low amplitude and within the alpha band (8–13 Hz). Therefore, it could represent the real alpha rhythm of EEG. From Figs. 8B and 8D, it can be seen that the SSA 2# could not extract any alpha rhythm in the eyes-open state. It was because the SSA 2# could only

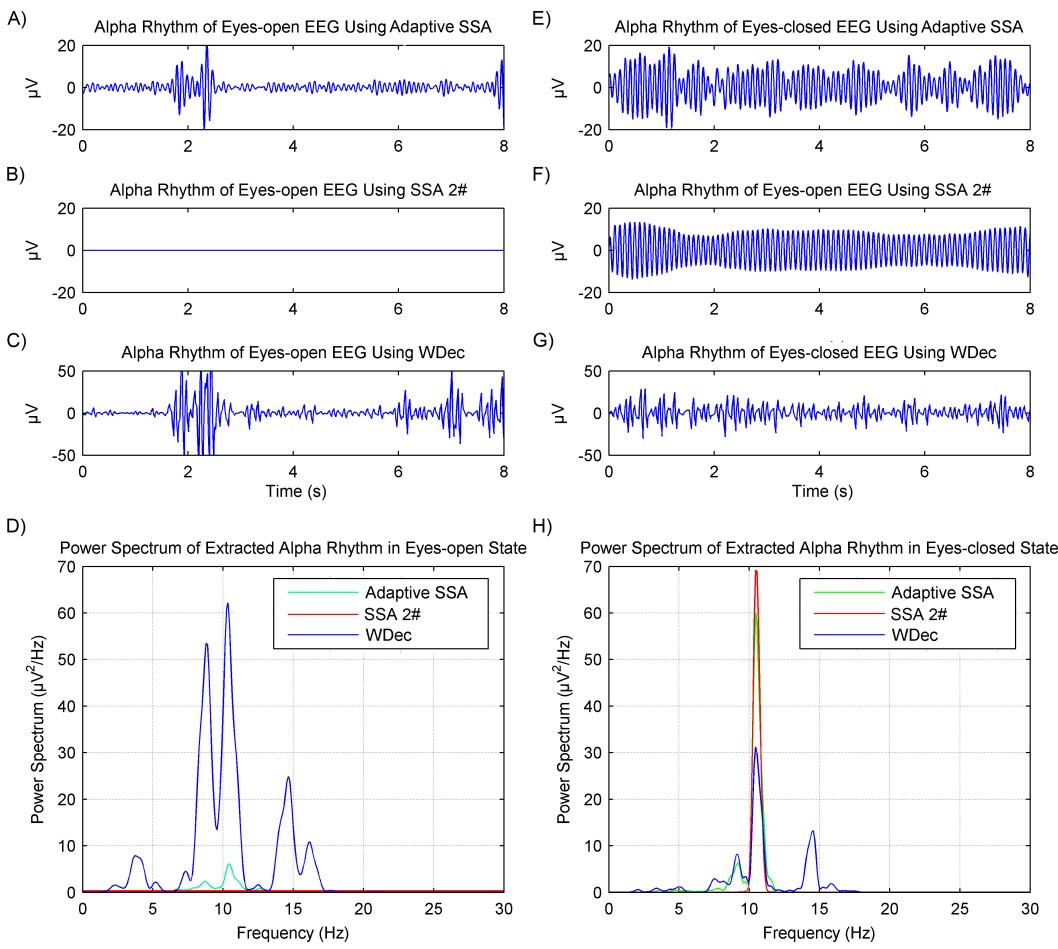

**Figure 8 Results of alpha rhythm extraction in the eyes-open and eyes-closed states using three different methods.** Extracted alpha rhythm using the proposed adaptive SSA (A), the SSA 2# (B) and the WDec (C), and power spectrums of the extracted alpha rhythms (D), in the eyes-open state. Extracted alpha rhythm using the proposed adaptive SSA (E), the SSA 2# (F) and the WDec (G), and power spectrums of the extracted alpha rhythms (H), in the eyes-closed state.

extract the dominate component from EEG; while the alpha rhythm in the eyes-open state is weak and not the dominate component. Figures 8C and 7D both show that the amplitude of the extracted alpha rhythm using the WDec in the eyes-open state was even higher than that in the eyes-closed state. It was obviously inconsistent with reality. Besides, it contained a large number of components out of the alpha band. Therefore, in the eyes-open state, the adaptive SSA performed better than both the SSA 2# and WDec in rhythms extraction.

In the eyes-closed state, the extracted alpha rhythms using the adaptive SSA and SSA 2# were both of high amplitude and within the alpha band. However, the extracted alpha rhythm using the WDec contained components out of the alpha band. Therefore, in the eyes-closed state, the adaptive SSA performed as well as the SSA 2#, but better than the WDec, in rhythms extraction.
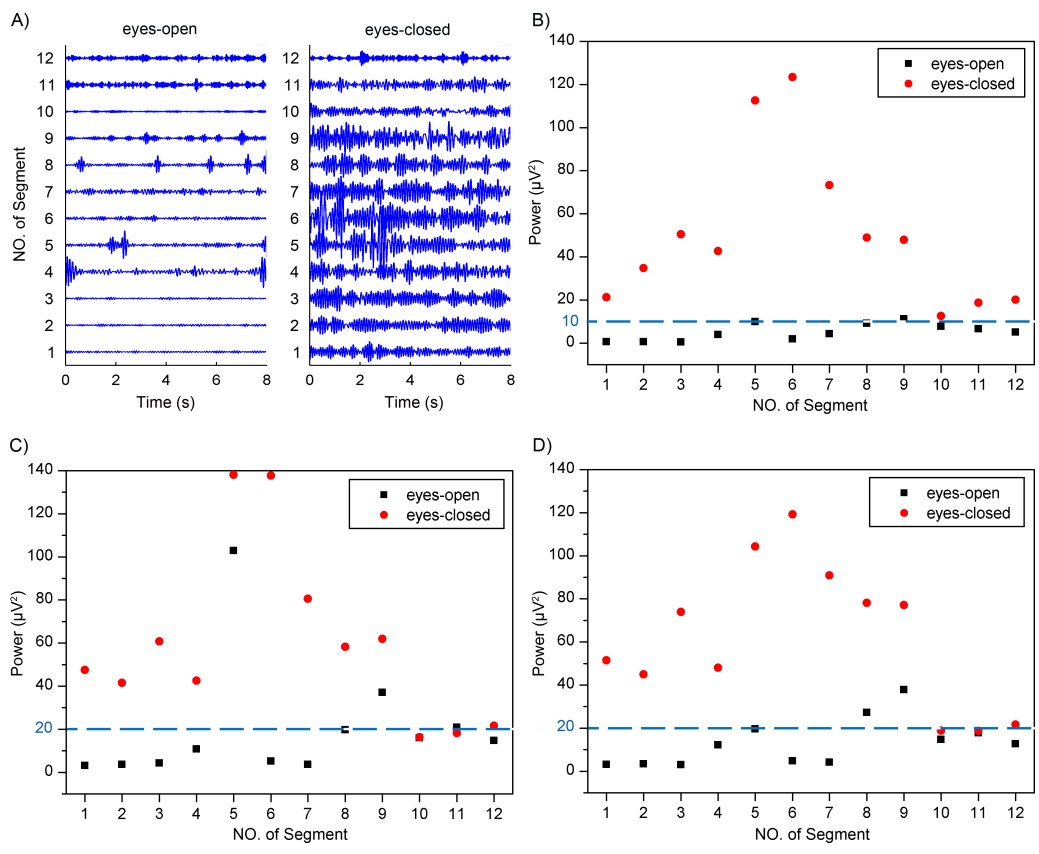

**Figure 9** (A) Extracted alpha rhythms using the adaptive SSA. (B) Power values of alpha rhythms using the adaptive SSA. (C) Power values of alpha rhythms using the WDec. (D) Power values of alpha rhythms using the IIR filtering.

## Distinguishment between the eyes-open and eyes-closed states

Previous studies have reported that the alpha rhythm in resting state with eyes closed is much stronger than that in the eyes-closed state with visual stimulation (*Barry et al., 2007*; *Norton et al., 2015*). Therefore, based on the alpha rhythm extraction, the adaptive SSA could be used to distinguish between the eyes-open and eyes-closed states. The extracted EEG alpha rhythms using the adaptive SSA are shown in Fig. 9A. In order to quantify the extracted alpha rhythms, their power values were calculated and shown in Fig. 9B. It is obvious that the power values in the eyes-open state were generally lower than those in the eyes-closed state. With a threshold power value of 10 $\mu V^2$, the two states could be distinguished. When the power was lower than the threshold, it was categorized as the eyes-open state. Otherwise, it was categorized as the eyes-closed state. The accuracy of the states distinguishment was 95.8%.

The performance of the adaptive SSA was compared with the WDec and the infinite impulse response (IIR) filtering methods. Figures 9C and 9D shows power values of the extracted alpha rhythms using the WDec and IIR filtering, respectively. It can be seen that power values in the eyes-open state were generally lower than those in the eyes-closed state, which was similar to the results of the adaptive SSA. However, the power values obtained

using the WDec and IIR filtering were larger than those using the adaptive SSA. It was likely because the WDec and IIR filtering could not remove the artifacts within the alpha band. With a threshold power value of 20 $\mu V^2$, the WDec and IIR filtering could both achieve the optimal distinguishment. The accuracy of states distinguishment was 79.2% and 83.3%, respectively, lower than the adaptive SSA. It can be concluded that the adaptive SSA could be potentially used to distinguish between the eyes-open and eyes-closed states.

## CONCLUSIONS

In this paper, we proposed an adaptive SSA method with a novel grouping rule to remove artifacts and extract alpha rhythms from EEG signals in eyes-open and eyes-closed states. The grouping rule enables SSA to be adaptive to EEG signals containing different levels of artifacts and rhythms. In order to verify the validity of the proposed adaptive SSA, the simulated EEG data based on the Markov Process Amplitude (MPA) EEG model and the experimental EEG data in eyes-open and eyes-closed states were used. The proposed adaptive SSA showed a better performance in artifacts removal and rhythms extraction, than another two recently reported SSA methods and the WDec method. Additionally, a proof-of-concept experiment was performed to apply the adaptive SSA to distinguish between the eyes-open and eyes-closed states. Results showed an accuracy of 95.8%, higher than that of the WDec method (79.2%) and the IIR filtering method (83.3%).

## APPENDIX

### Markov Process Amplitude (MPA) EEG model

As described in 'Artifacts removal', the EEG is a combination of several oscillations. The MPA EEG model can be constructed as

$$
\begin{cases}
s(n\Delta t) = \sum_{i=1}^{K} a_i(n\Delta t)\sin(2\pi m_i n\Delta t + \theta_i) \\
a_i[(n+1)\Delta t] = \gamma_i a_i(n\Delta t) + \xi_i(n\Delta t) \\
0 < \gamma_i < 1 \qquad i = 1, 2, \ldots, K
\end{cases}
\tag{8}
$$

The theoretical power spectrum of the MPA EEG model was given as

$$
P(f) = \sum_{i=1}^{K} \left\{ \frac{0.25\Delta t \left(\sigma_i^{\xi}\right)^2}{1 + (\gamma_i)^2 - 2\gamma_i \cos[2\pi \Delta t \left(f - m_i\right)]} + \frac{0.25\Delta t \left(\sigma_i^{\xi}\right)^2}{1 + (\gamma_i)^2 - 2\gamma_i \cos[2\pi \Delta t \left(f + m_i\right)]} \right\}.
\tag{9}
$$

In the power spectrum of the EEG, the width and the amplitude of peak frequency are the most important features. Define $H_i$ as the amplitude and the $F_i$ as frequency width at half of $H_i$. $H_i$, $F_i$ can be described as

$$
H_i = \frac{\Delta t \left(\sigma_i^{\xi}\right)^2}{4(1 - \gamma_i)^2}
\tag{10}
$$

$$
F_i = \frac{1}{\pi \Delta t} \cos^{-1} \frac{4\gamma_i - 1 - (\gamma_i)^2}{2\gamma_i}.
\tag{11}
$$

In order to determine parameters of the MPA EEG model, the power spectrum of the real EEG was calculated using the Welch method. The amplitude and width of peak frequency ($m_i$) in deferent oscillations (delta, theta, alpha and beta) were obtained as $H_i$ and $F_i$. Then, the parameters of the model, $\gamma_i$ and $\sigma_i^{\xi}$, were determined.

### Funding

This work was supported by the Tsinghua University Initiative Scientific Research Program (20131089190). The funders had no role in study design, data collection and analysis, decision to publish, or preparation of the manuscript.

### Grant Disclosures

The following grant information was disclosed by the authors:
Tsinghua University Initiative Scientific Research Program: 20131089190.

### Competing Interests

The authors declare there are no competing interests.

### Author Contributions

- Hai Hu conceived and designed the experiments, performed the experiments, analyzed the data, contributed reagents/materials/analysis tools, wrote the paper, prepared figures and/or tables, reviewed drafts of the paper.
- Shengxin Guo performed the experiments, analyzed the data, reviewed drafts of the paper.
- Ran Liu contributed reagents/materials/analysis tools, reviewed drafts of the paper.
- Peng Wang analyzed the data, wrote the paper, reviewed drafts of the paper.

### Human Ethics

The following information was supplied relating to ethical approvals (i.e., approving body and any reference numbers):

The research was approved by the Institutional Review Board (IRB) of Tsinghua University (Ethical Application No.: 20170010).

### Data Availability

The raw data has been supplied as a Supplementary File.

### Supplemental Information

Supplemental information for this article can be found online at http://dx.doi.org/10.7717/peerj.3474#supplemental-information.

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
