# Peer review of "An adaptive singular spectrum analysis method for extracting brain rhythms of electroencephalography"

_PeerJ, doi:10.7717/peerj.3474_

## Round 0.1 · original submission · Major Revisions

· Academic Editor

Major Revisions

The reviewers have identified major issues that need to be addressed in any revised manuscript.

The novelty of the method is not made sufficiently clear. The authors need to properly reference parts of the methods which have previously been proposed and describe the novel contribution of the current paper (the adaptive grouping rule) in more detail. Recent alternative approaches should be discussed and used as reference to compare to the proposed method. A more realistic computational model should be used to simulate EEG activity and preferably publicly available EEG should be used as experimental EEG used for the comparison.

Reviewer 1 ·

Basic reporting

1. The paper is not well organized and written. The paper need a thoroughly proof reading. For example, the "method" in abstraction:
What is your "cleverly selected grouping rule"? How "the adaptive SSA method could select the subspace of the desired signal components automatically"? What does it mean " Simulated and experimental single-channel EEG data was collected to perform the method"?
2. Literature review lacks of many recent references on EEG analysis using SSA, for example,
1). Improving time–frequency domain sleep EEG classification via singular spectrum analysis,
2.) Motion artifact removal from single channel electroencephalogram signals using singular spectrum analysis
3.) Investigation of the noise effect on fractal dimension of EEG in schizophrenia patients using wavelet and SSA-based approaches

Experimental design

SSA is well known and the authors should focus on describing your adaptive grouping rule with sufficient detail.

Validity of the findings

There lacks comparison of the proposed method with those up-to-date methods I list. And the experimental research is too weak to support their findings, especially using the simple threshold in Fig. 5.

Reviewer 2 ·

Basic reporting

The manuscript has not been written in a scientific English language using clear and unambiguous text. The article does not propose a new method or an idea in order to extract brain rhythms. There is no sufficient introduction and background to demonstrate how the work fits into the broader field of brain signal processing. For example, the time-delay embedding approach used by the authors of this manuscript is called Takens' theorem (F. Takens (1981). "Detecting strange attractors in turbulence". In D. A. Rand and L.-S. Young. Dynamical Systems and Turbulence, Lecture Notes in Mathematics, vol. 898. Springer-Verlag. pp. 366–381). The authors did not mention the original name of this technique, and hence confused the reviewer. The application of SVD in the embedded time-series is not also a new idea (https://journals.aps.org/pra/abstract/10.1103/PhysRevA.36.340). The mathematics and figures of the manuscript are not professional and precise. Furthermore, the use of Fast Fourier Transform (FFT) in the estimation of a long-term EEG signal is absolutely wrong, please REVISE your experimental and method sections.

Experimental design

The experimental design is not clear. The research question in this manuscript may be of interest, but the computational and theoretical design were not formulated accurately and professionally. This must be addressed in a clear fashion. The use of the Gaussian noise with sinusoidal functions at different frequencies are the basic (and weak) simulation study - use a more complex model to simulate EEG.

Validity of the findings

It's not straightforward to validate these findings, since the Matlab codes/functions are not precisely matched with the weak description of SVD matrix formulation given in the manuscript.

Reviewer 3 ·

Basic reporting

I do not feel qualified to comment upon the English language. However, I think that the language used is clear and unambiguous. There are sufficient references to frame this work in its contest. The paper is self contained, results are relevant to the stated hypotheses. Furthermore they can be verified from the shared data.
The figures may need some work to improve readability. I suggest to make the font for axis and legends bigger and orient large figures like Fig 3 and 4 horizontally so they are larger and easier to see/zoom in particularly where color-bars are used

Experimental design

The work is withing the Aims and Scope of the journal. The research question is relevant to the field and addressed properly. This work may benefit from an extension of the method to publicly available EEG datasets. However, the presence of the recorded data is technically sufficient at this stage.

Validity of the findings

Conclusions are well stated and relevant to the project proposed. As I said above remain interesting to see how the algorithm performs on other EEG datasets.

Comments for the author

no comments to add

---

## Round 0.2 · accepted · Accept

· Academic Editor

Accept

The authors have revised their manuscript considerably. The novel aspects of the method has been clarified and the proposed method is now compared with the wavelet decomposition method and another two recently reported SSA methods. The major concerns expressed by the reviewers have been addressed.

Reviewer 1 ·

Basic reporting

The paper has been significantly improved and my major concerns have been clarified in the revision.

Experimental design

revision: no further comments

Validity of the findings

revision: no further comments

Comments for the author

revision: no further comments